# Utilization of social health security scheme among the households of Illam district, Nepal

**Sanjeeb Shah**[1]*, **Nilambar Jha**[1], **Vijay Kumar Khanal**[1], **Gyanu Nepal Gurung**[1], **Babita Sharma**[2], **Mausam Shrestha**[2]

1 School of Public Health and Community Medicine, B.P. Koirala Institute of Health Sciences, Dharan, Nepal,
2 Karuna Foundation Nepal, Biratnagar, Morang, Nepal

* sanjeeb619shah@gmail.com

**Data Availability Statement:** All relevant data are within the paper and its supporting information files.

## Abstract

### Background

Nepal has implemented the social health security program in certain health facilities to improve access to quality health services. The aim of the study is to understand the utilization pattern of social health security schemes and community's perception of the social health security program.

### Method

A descriptive cross-sectional mixed-method study was conducted among 300 households of Illam district who had enrolled in the social health security scheme. A multi-stage random sampling method was used. A semi-structured questionnaire was used to collect quantitative data whereas focus group discussions (FGDs) were conducted for qualitative data. Descriptive analysis, bivariate and multivariate analysis was done. FGDs were transcribed and thematic analysis was done.

### Findings

The utilization rate of social health security scheme was 88.7%. Factors associated with the utilization of program included: presence of under-five children and chronic illness in households, sex and age group. Focus group discussion revealed that people were aware of social health insurance and pleased with program implemented by government. However, the hospitals under the social health insurance were not able to meet their expectations.

### Conclusion

Although the utilization of social health security scheme was high, people were dissatisfied with the service provided by the hospital under the social health security scheme. Therefore, the social health insurance board should closely monitor the hospitals and develop a feedback mechanism from the users.

**Funding:** The author SS was awarded by Nepal Health Research Council, Post Graduate Health Research Grant with reference number 2376. http://nhrc.gov.np/ The funder had no role in study design, data collection and analysis, decision to publish, or preparation of the manuscript

**Competing interests:** The authors have declared that no competing interests exist.

## Introduction

The World Health Assembly appealed to the countries to achieve Universal Health Coverage in 2005. However, it is uncertain how this ambitious goal can be reached [1]. One way to achieve this is by reducing inequities. Social health insurance is a health care financing mechanism that is based on comprehensive social contributory scheme where the government provides subsidies for the poor. It is believed to be a powerful instrument for eliminating unmet health needs [2].

Although low-income countries account for half a billion people, only 1% of the world health spending is allocated to those countries [3]. Moreover, in low and middle-income countries (LMIC), most of the domestic health care expenditure is out-of-pocket payment rather than prepaid insurance [4]. In Sub-Saharan Africa and Southern Asia only 5–10% of people are covered with social security programs, while in middle-income countries coverage rates range from 20% to 60% [5].

Nepal remains committed to achieving universal health coverage. Although Nepal has a wide network of health facilities, only two-thirds of the populations have easy access to health care [6]. Despite various treatment subsidies like incentives program and pilot insurance scheme, out-of-pocket payment (OOP) is still the main way to finance health care accounting for 48% of total health expenditure [7]. Thus, there was a felt need for a mechanism in the health financing system to enhance prepayment and risk pooling to minimise financial hardship and impoverishment arising from the use of health care services in Nepal.

The government of Nepal introduced the social health insurance program to government run health facility (Primary Health Care Centre, District Hospital which are reachable within one hours) that provide out-patient and in-patient service to mitigate the financial hardship due to healthcare and increase access to healthcare. The program was piloted in 2016 in three districts: Kailali, Baglung and Illam. The Social Health Security Programme (SHSP) is a social protection programme of the Government of Nepal that aims to enable its citizens to access quality health care services without placing a financial burden on them. In 2015 Government of Nepal established a semi-autonomous body, Social Health Security Development Committee which is now known as Health Insurance Board as an implementer of program which follows the structure of centre, province, district and ward. For a yearly membership fee (Premium) of Rs. 2500 ($22), a family of up-to five member (4.2 average family size of Nepal [8]) could receive an insurance card and an annual coverage of benefit package up to Rs. 50000 ($441) health related expenses in both public and private social health insurance empanelled hospitals. The scheme includes the benefit package of emergency services, out-patient services, selected inpatient services, selected diagnostic services and selected drugs, in addition to any free services and drugs available at public health facilities through other programmes. The government provides subsidies based on wealth i.e. for ultra-poor 100%, Poor 75% and vulnerable groups 50% discount in premium. The choice of enrolment was voluntary. This cashless scheme covered pre-existing conditions and had no age limits [2]. The coverage rate was 5% with a catchment population of 2, 38,344 people across the 15 implemented districts [9].

Thus a comprehensive exploration of social health security scheme was needed as rigorous research evidence is lacking regarding the utilization of social health security scheme. This study therefore aims to assess the utilization pattern of social health security scheme and explore the community's perception of social health security/insurance program.

## Methods

### Study design

A descriptive cross-sectional method was adopted with a mixed method approach to study the 300 households that were enrolled in the health insurance scheme of Illam district of Nepal between September 2018 –February 2019 and whose date of service had come into effect.

### Sample size and sample selection

A sample size of 300 was calculated using a formula $n = (z^2pq)/e^2$ considering the prevalence from study done in 2016 by Philip NE. et al. in Kerala India and non-response rate of 10%.

A 4 stage random sampling technique (district, municipality, ward and household) was used to select the sample through population proportionate method.

**Selection of district.** Government of Nepal initiated the Social Health Security program in 2016 AD and piloted in three districts (Kailali, Baglung and Illam) from which Illam district was selected randomly through lottery method.

**Selection of municipality.** Out of six rural municipalities, two rural municipalities (Maijogmai and Chulachuli) and out of four urban municipalities, one urban municipality (Illam) was selected through lottery method for the equal representation.

**Selection of ward.** Three wards from each of the selected rural municipality and urban municipality was then chosen through lottery method and the sample was taken via population proportionate method.

**Selection of household.** Lastly household was selected randomly by spin the pen technique [10] (a pen was spin and every household who was insured and whose date of service has come into effect/activated along the line in the direction was approached until the required sample size was achieved). If the required sample size was not achieved from the earlier direction, spin the pen technique was repeated and households in the other direction were approached till the required sample size was achieved. For this map was generated manually with support of social health insurance supervisor of Illam district based on data available there.

### Data collection

**Quantitative.** Data was collected by a researcher using pretested semi-structured questionnaires through face to face interview developed through intensive literature review according to research objective and methodology. Any household member more than 18 years of age was interviewed to obtain the required information. Information on utilization of In-patient and Out-patient and other schemes was obtained for 12 months before the date of interview.

**Qualitative.** The principle of phenomenology was used to conduct three focus group discussions (FGD) in selected three municipalities (one in each municipality) using pretested focus group guideline prepared based on the theme. Each focus group discussion lasted for 30–40 minutes which included 10 participants chose purposively from insured household, whose date of service was into effect, ensuring representation of gender, different castes, religious group and elderly. Note taking as well as audio recording of discussion was done after taking consent from participants. The purpose of FGD was to record the community's perception of social health security scheme.

### Data analysis

Collected data was entered in Microsoft Excel 2010 and was analyzed in Statistical Package for Social Science 11.5. Descriptive, bivariate and multivariate analysis was done. Bivariate analysis

was done using chi-square test to establish relationship between dependent (utilization of social health security scheme) and independent (household characteristics) variables. The level of significance was set at 5%. Variables with p-value<0.2 and cell count more than 5 in the bivariate analysis were included for the binary logistic regression analysis [11] to find the factors associated with utilization of social health security scheme.

For qualitative, data was coded, transcribed and thematic analysis [12] was performed manually where themes were identified in advance.

### Ethical consideration

Ethical clearance was obtained from the Institutional Review Committee (IRC) of BPKIHS (Code No. IRC/1275/018). Informed written consent was obtained from the respondents. Consent was taken from all the co-authors for the publication.

## Result

The quantitative results are presented followed by qualitative.

### Quantitative results

A total of 300 households with 1421 members were included in the study.

**Household characteristics of respondents.** Out of 300 households, more than half of households (56%) were Brahmin/Chhetri (Brahmin/Chhetri are considered to be the upper caste group, Janjati and Dalit are considered to be the lower caste/disadvantage group in society) by ethnicity and 55% had nuclear families comprises of parents and children whereas other households were extended families including other relatives too. The mean household size was 4.7 with 25% having under five children and 32% aving an elderly member. Family member with chronic illness were present in 140 households (46.7%.). More than half of households (56.3%) had two or more working members in family (Table 1).

**Utilization of social health security scheme.** Table 2 shows that; the utilization rate of social health security scheme among the 300 households was 88.7%. A total of 266 households had utilized the social health security scheme, 23.1% utilized the scheme for fever followed by cardiovascular diseases (17.1%) and neurosensory disease (14.7%) respectively. Among the households not utilizing the social health security (n = 34), 100% gave the reason that the nearby private health facility was easy to access. A majority of the households (90.8%) used the social health security scheme for minor morbidity. The utilization rate of receiving drugs, outpatient service, laboratory service, in-patient service and emergency service was 96.2%, 89.4%, 67.7% 10.4% and 0.4% respectively. Around half of the households (49.2%) used the social health security scheme twice.

**Factors associated with utilization of social health security scheme (at household level).** Presence of under-five children and chronic illness was statistically significant with utilization of social health security scheme by insured household. Therefore, households with under five children had a 2.91 times greater odds of utilizing the social health security scheme in comparison to households not having under five children. Similarly, households reporting a chronic illness among a family member had a 3.83 times greater odds of utilizing scheme than households not reporting (Table 3).

**Factors associated with utilization of social health security scheme (at individual level).** Male members of household had 64% lower odds of utilizing the social health security scheme than females. Age group of 15–59 years had 74% lower odds of utilizing the social health security scheme in comparison to age group 60 years or above (Table 4).

**Table 1. Household characteristics of respondents (n = 300).**

| Characteristics | Frequency | Percent (%) |
|---|---|---|
| **Ethnicity** | | |
| Brahmin/Chhetri* | 168 | 56 |
| Janajati | 121 | 40.3 |
| Dalit | 11 | 3.7 |
| **Family type** | | |
| Nuclear# | 165 | 55 |
| Extended | 135 | 45 |
| **Household size** | | |
| < = 5 | 225 | 75 |
| >5 | 75 | 25 |
| Mean (SD) = 4.74 (1.44) | | |
| **Presence of under-five children** | | |
| No | 218 | 72.7 |
| Yes | 82 | 27.3 |
| **Presence of elderly member(>60)** | | |
| No | 203 | 67.7 |
| Yes | 97 | 32.3 |
| **Presence of chronic illness** | | |
| No | 160 | 53.3 |
| Yes | 140 | 46.7 |
| **No. of working member in household** | | |
| ≥2 | 169 | 56.3 |
| 1 | 131 | 43.7 |
| **Poverty line of household** | | |
| Above poverty line | 158 | 52.7 |
| Below poverty line | 142 | 47.3 |

Note: Poverty line was calculated according to World Bank standard i.e less than USD 1.90 per day head as below poverty line and more than USD 1.90 per day per head as above poverty line. (1 USD = 113.38 NPR as per 2019/01/18)

*Brahim/Chhetri-upper caste group, Janjati and Dalit Lower caste/Disadvantage group in society

#Nuclear family comprise of parents and children and extended family includes other relatives

## Qualitative result

The themes developed for analysis related to community's perception on social health insurance included;

- Social health insurance related knowledge

- Experience of getting enrolled in social health insurance and

- Experience on the utilization of social health insurance service.

**Social health insurance related knowledge.** The participants stated that social health insurance is a Nepal Government program to provide health services for people. They stressed it was in particular for poor people and to overcome unpredicted health problem during an economic crisis. This information was received from the Media- radio, television, FCHVs, enrollment assistant (who register for the health insurance).

All of the participants were aware of premium amount i.e. Rs.2500 for a five family member and Rs.425 for any additional family member. They suggested lessening the premium amount than the existing sum.

**Table 2. Utilization of social health security scheme (n = 300).**

| Characteristics | Frequency | Percent (%) |
|---|---|---|
| **Household utilized the social health security scheme** | | |
| Yes | 266 | 88.7 |
| No | 34 | 11.3 |
| **Reason for using social health security scheme (n = 266)** | | |
| Fever | 115 | 23.1 |
| Cardiovascular disease | 85 | 17.1 |
| Neurosensory disease | 73 | 14.7 |
| Musculoskeletal disease | 62 | 12.5 |
| Endocrinal disease | 56 | 11.3 |
| Gastrointestinal disease | 47 | 9.5 |
| Respiratory disease | 41 | 8.2 |
| Reproductive disease | 33 | 6.6 |
| Urinary disease | 18 | 3.6 |
| Dental disease | 16 | 3.2 |
| **Reason for not using social health security scheme (n = 34)** | | |
| Nearby health service are easy to access | 34 | 100 |
| **Social health security scheme used for type of morbidity (n = 266)** | | |
| Minor morbidity | 452 | 90.8 |
| Major morbidity | 51 | 10.2 |
| **Type of Social health security scheme utilized by household (n = 266)** | | |
| Drugs | 483 | 96.2 |
| Out-patient service | 449 | 89.4 |
| Lab diagnostic service | 340 | 67.7 |
| In-patient service | 52 | 10.4 |
| Emergency service | 2 | 0.4 |
| **No. of time household member used social health security scheme (n = 266)** | | |
| 2 times | 131 | 49.2 |
| 1 time | 70 | 26.3 |
| 3 time | 53 | 19.9 |
| 4 = > times | 12 | 4.6 |

**Table 3. Factors associated with utilization of social health security scheme (at household level).**

| Characteristics | Adjusted Odds Ratio # | 95%CI | P-value |
|---|---|---|---|
| **Ethnicity** | | | |
| Dalit | 2.644 | 0.582–12.019 | 0.208 |
| Janajati | 1.375 | 0.630–3.001 | 0.423 |
| Brahmin/Chhetri | Reference | | |
| **Family type** | | | |
| Nuclear | 0.420 | 0.600–3.863 | 0.377 |
| Extended | Reference | | |
| **Presence of Under five children** | | | |
| Yes | 2.915 | 1.324–6.419 | **0.008*** |
| No | Reference | | |
| **Presence of chronic illness** | | | |
| Yes | 3.830 | 1.422–10.316 | **0.008*** |
| No | Reference | | |
| **No. of working member in household** | | | |
| 1 | 1.067 | 0.464–2.453 | 0.879 |
| ≥2 | Reference | | |

"*" Significant Association ($P < 0.05$)

"#" Adjusted for ethnicity, family type, presence of under five children, presence of chronic illness, no. of working member in household

**Table 4. Factors associated with utilization of social health security scheme (at individual level).**

| Characteristics | Adjusted Odds Ratio # | 95%CI | P-value |
|---|---:|---:|---:|
| **Sex** | | | |
| Male | 0.563 | 0.448–0.707 | **<0.001*** |
| Female | Reference | | |
| **Age group** | | | |
| ≤14 | 0.991 | 0.730–1.345 | 0.955 |
| 15–59 | 0.263 | 0.182–0.381 | **<0.001*** |
| ≥60 | Reference | | |

"*" Significant Association (P < 0.05)

"#" Adjusted for sex and age group.

Most of them were unaware about the subsidy provided by social health insurance board for poor, ultra-poor and marginalized people. Most of participants shared that they can get drugs, video x-ray (ultra-sound), chest x-ray services through social health insurance. "None of them knew about which services were provided through the scheme". The reason for this was; people were not explained about what services were available as part of scheme nor given any service list booklet during the enrollment process.

**Experience of getting enrolled in social health insurance.** Participants stated that the reason for getting enrolled into the health insurance scheme was to protect the family from unpredicted financial risk when they got a health problem. They also mentioned lack of cash at their home for medical emergencies. Almost all the participants stated that it was easy for them to get enrolled in social health insurance because enrollment assistant was visiting their house for enrollment procedure on regular basis. "*Able to receive health service like video x-ray (ultra-sound) for kidney stone regularly, drugs, health check up by simply showing the health insurance card at the time when we do not have money is huge benefit for us, having something is better than having nothing*" (-female).

**Experience on social health insurance service utilization.** Everyone shared that "the social health insurance is very good program introduced by Government of Nepal to make the health service accessible to every citizen". But it's difficult to get services when we go to hospital as we do not find the "doctor in hospital", "all kind of diagnostic service", "drugs" and also "people with health insurance in hospital has to go through many registrations counters". People were unsatisfied by the biased treatment from hospital staff between insured and uninsured, misbehavior in pharmacy and some had to go back home without receiving their service. In addition, participants did not find a good referral system or the doctor did not give a referral until the condition got worse. We have to go either Biratnagar or Dharan with referral. This puts extra economic burden. "*He went for cyst checkup in hospital, at first when hospital staff did not know that he had health insurance the staff asked him to wait as doctor will come at 2 pm. So he waited and after sometime he said to hospital staff that he has health insurance then the staff said that doctor is on leave and will come after 2–3 days only* (-Male). Then thereafter went to a private clinic.

People suggested that everyone should be treated equally at the hospital irrespective of insurance status. Hospital should also have all the covered facilities and services listed for the patients. This might decrease referral and cost of transport. Most of the people stated that they should be able to use the health insurance card in any place, at any hospital under social health insurance without going through the referral procedure. All participants shared that the service should start the day they get enrolled. The program should be implemented in all the private hospitals that have taken permission from the government.

## Discussion

### Utilization of social health security scheme

Limited studies have been conducted on the social health insurance program in the context of Nepal. The study revealed higher utilization of social health security scheme which is consistent with the studies done in Philippines and Rwanda [13, 14] However, India has a lower finding [15]. The utilization of outpatient service was higher than inpatient service which aligned with the finding from South Africa, Kenya, Senegal and contrast to India and China [16, 17, 18] Lesser utilization of laboratory service was explicit in study conducted in Chile than in this study [19]. The underutilization of the scheme was slightly lower than that of study from Philippines [13]. The reason for this might be due to easy access to private health services.

### Factors associated with utilization of social health security scheme

The study depicted that households having under five children and chronic illness were more likely to utilize the social health security scheme, similar to India's finding [17] suggesting expected phenomenon of voluntary health insurance scheme that there is likely to be adverse selection from an insurer side. In contrast, to study done in South Africa, Kenya and India sex has made difference on scheme utilization in this study. Regarding age, older age group (i.e. 60 and above years of people were more likely to utilize the social health security scheme which is similar with to finding from South Africa and China [16, 18].

### Community's perception

The FGD enlightened that people had understood what social health insurance program is all about. They came to know about it through radio, television, FCHVs and enrollment assistant. However, none of them exactly knew about what services are provided through social health insurance in hospital and most of them were unaware about the subsidize provided by social health insurance board for poor, ultra-poor and marginalized people as the insurance board was unable to implement the policy because government has not distributed the poverty card in Illam district.

Irrespective of people different financial status, they had enrolled in social health insurance because they had the concept that health insurance protects their family from unpredicted financial risk when they get health problem. In addition to that it was easy for people to get enrolled in social health insurance because for each ward one enrollment assistant has been appointed.

People supported the social health insurance program but they had unpleasant experience while taking service at hospital. They faced discrimination with uninsured people, misbehaviors at hospital pharmacy, have to go through numbers of registration counters, did not find the doctors, all kind of lab services, drugs, and difficulty with referral system. Therefore, for the further improvement of the social health insurance program, people suggested to treat everyone equally in hospital, all the services should be available in nearby hospital, should be able to use insurance card at any place, at any time, at any hospital under social health insurance without going through referral procedure. The qualitative findings (FGD) from the study done in India shared that Rashtriya Swasthya Bima Yojana (National Health Insurance Scheme) the people were unaware about the what services are provided by the Scheme and from which hospital they can get the service. Moreover, insurance did not reached the poor people and discrimination was done based on political power and insurer were compel to make additional payment and pressurized to buy medicine from out of pocket [20].

## Limitations

There might have occurred recall bias as the recall period of utilization of service was 12 month before the interview. However, it was minimized with medical bills and report wherever provided but was not applicable in each household. The study could not cover the uninsured population. Therefore who were more likely to enrol than others, reason or barrier for not enrolling, what kind of facilities accessed by whom or utilization of service by different provider, what is the difference in out of pocket spending remained uncovered. However this can be examined in further studies.

## Conclusion and recommendation

In Nepal with increased focus on achieving universal health coverage through increasing access to health care, the study provides intuition to which extent this dimension of universal health coverage has been achieved through social health security scheme program in Illam district. Although the utilization of social health security scheme was higher and had positive response towards the program, people were unsatisfied with the service and facility provided by the hospital under the social health security scheme. Therefore, the investment of demand-side will not be justified unless the basic health care infrastructures are upgraded to specialist health besides serving as gatekeeping for specialist service. In addition to that the concern authority should invest in system to monitor the hospitals and develop a feedback mechanism from users so to evaluate the implementation of health insurance scheme. This will further help the government to refine the policy, plan and strategies in addressing the factors that hindered the consumers' satisfaction. Moreover, such system will provide robust path in achieving the government's goal of scaling up the program nationwide. Further, a study from provider perspective would give more clear insight on implementation situation of social health security program.

## Supporting information

**S1 Checklist.**
(DOCX)

**S1 Appendix.**
(DOCX)

**S1 Data.**
(SAV)

## Acknowledgments

Author would like to express deepest gratitude to BPKIHS, the participants consenting to get enrolled in the study, Mr. Sandesh Rajthala, Mr. Anil sigdel and Mr. Khem Raj Adhikari and his team of social health insurance board Illam district.

## Author Contributions

**Conceptualization:** Sanjeeb Shah.

**Data curation:** Sanjeeb Shah, Mausam Shrestha.

**Formal analysis:** Sanjeeb Shah.

**Funding acquisition:** Sanjeeb Shah.

**Methodology:** Sanjeeb Shah, Nilambar Jha, Vijay Kumar Khanal, Gyanu Nepal Gurung.

**Software:** Sanjeeb Shah.

**Supervision:** Nilambar Jha, Vijay Kumar Khanal, Gyanu Nepal Gurung.

**Validation:** Sanjeeb Shah.

**Writing – original draft:** Sanjeeb Shah, Babita Sharma.

**Writing – review & editing:** Sanjeeb Shah, Mausam Shrestha.

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
