## [Decision Letter · Decision Letter 0]

20 Oct 2020

PONE-D-20-13306

Utilization of social health security scheme among the households of Illam district, Nepal

PLOS ONE

Dear Dr. Shah,

Thank you for submitting your manuscript to PLOS ONE. After careful consideration, we feel that it has merit but does not fully meet PLOS ONE’s publication criteria as it currently stands. Therefore, we invite you to submit a revised version of the manuscript that addresses the points raised during the review process.

We look forward to receiving your revised manuscript.

Kind regards,

Nagendra P. Luitel, MA

Academic Editor

PLOS ONE

Journal Requirements:

3.We suggest you thoroughly copyedit your manuscript for language usage, spelling, and grammar. If you do not know anyone who can help you do this, you may wish to consider employing a professional scientific editing service.  

Reviewers' comments:

Reviewer's Responses to Questions

**Comments to the Author**

1. Is the manuscript technically sound, and do the data support the conclusions?

Reviewer #1: No

Reviewer #2: Partly

2. Has the statistical analysis been performed appropriately and rigorously? 

Reviewer #1: No

Reviewer #2: Yes

3. Have the authors made all data underlying the findings in their manuscript fully available?

Reviewer #1: No

Reviewer #2: Yes

4. Is the manuscript presented in an intelligible fashion and written in standard English?

Reviewer #1: No

Reviewer #2: No

5. Review Comments to the Author

Reviewer #1: Manuscript Title: Utilisation of social health security scheme among the households of Illam district, Nepal

Manuscript Number: PONE-D-20-13306

The Govt. of Nepal has been implementing a Social Health Insurance program, and this particular research aims to assess utilization patterns of the scheme in one district, Illam, Nepal. While it is critical to conduct such exercise given its importance of being used as inputs for monitoring and revising the scheme design, following observations may be useful in respect to the manuscript in its current form.

The manuscript hardly provides any idea about the background of social security program. How many people are covered, which sections of population are covered, are there targeted mechanism? What is the benefit package offered by the scheme, and is this scheme purely health security or a larger social security scheme? and if so, what aspect of the survey captured in the manuscript? What kind of purchasing mechanism did the insurance scheme implement – did they empanel both public and private hospitals? A reading of the text later suggests that the manuscript deals with social health security and not social security scheme at large. Moreover, further down the text in sections dealing with qualitative results, the authors appear to use the term social health insurance. The manuscript must therefore clearly articulate what type of scheme are they referring to?

Further questions underlying the current manuscript are organized in the following paragraphs:

Study design:

1. Why was Illam district selected as against other districts? ON what basis the samples were selected? Is that proportionate to social security coverage? The sample size calculation is utterly confusing, the method is unclear. A sample of 300 was collected based on Inpatient utilization of 38.5% (this is unrealistic given that more than one of the population of the district requires hospitalization in a year), which is usually 4-6% in most LMIC countries?

2. Why choose six months rather than one year as data period. If this is for hospitalization did the survey collect information of self-reports for one year for hospitalization or 15/30 days for OP? The choice of recall period for OP and IP is not spelt out in the manuscript.

3. The method of selection of household was even more bizarre. Not sure what spinning the pen techniques is all about? A reference to this technique must be provided, if such a technique is in vogue or were practiced earlier?

4. The qualitative method – no information is available in the text about what sets of variables are part of the questionnaire. Besides socio-economic profile of households, what other information were obtained? Did the tools collect information about different illness, did they record utilization of services by different types of providers, and did the researcher collect information about households’ OOP and if so, what are the key variables involved? None of these information/details are provided in the paper.

5. From qualitative method, it was clear that the research was conducted only among those who were insured, leaving the uninsured out. It is equally possible that those who are eligible for insurance but uncovered for any reasons. And this set of population is expected to be significant, which is left out of the survey.

Data Analysis:

The data analysis section is presented in a vague manner for any meaningful understanding? Why were binary logistic regression analysis chosen. What is the rationale behind choosing this analysis technique? What does the manuscript wants to convey when it says it carried out thematic analysis underlying qualitative data?

Findings:

- For readers in other parts of the world may not be familiar with words such as Brahmin/Chhetri reflecting ethnicity in Nepal. Perhaps a note on these characteristics is required.

- It is unrealistic to expect nearly half of the population (46.7%) to suffer from chronic illness. Either the method adopted to capture illness (self-reports) or the sample technique adopted was skewed. Even assuming nearly half the population have chronic illness in a particular district, does that reflect national average, or global evidence?

- While chronic illness were present in 46.7% of households as reported in Table 1, but

also, it appears that about 10.4% are reported to be hospitalized, which looks extremely high.

- How did the survey go about classifying the disease conditions (broader disease classification) as given in Table 2. Is that based on ICD-10 or what kind of classification?

- The section on qualitative data dealing with social health insurance related knowledge, it is abundantly clear that the scheme is yet to be implemented (since the poverty card in Illam district has not been distributed yet). In the absence of implementation of the scheme in the sample district, it is unclear what kind of results can be expected from the survey? But the next paragraph provides a contradictory view of beneficiaries receiving insurance benefits.

The issues outlined above needs serious consideration before the manuscript can be published.

Reviewer #2: There have not been many studies on Nepal’s new health insurance scheme and any attempts to understand it are welcome. However, the current study has severe weaknesses which need to be addressed.

1. The insurance scheme details have not been provided – what are its stated objectives, what kind of people are eligible for insurance scheme, who pays the premium, what part of the population is eligible for waivers, what is the role of government, what kind of providers are used – public, private or both, what kind of care is included – inpatient/out-patient/both etc. If it is a voluntary insurance scheme, it should be mentioned upfront and compared with literature on voluntary insurance.

2. Any literature available on functioning of the scheme and its origin should be included.

3. It is not clear why the scheme was introduced. Though the introduction mentions that the context was of moving away from public sector provisioning, it is not backed by any references.

4. The main aspect the study is able to look at is utilisation of insurance scheme by those who were insured. It seems that the families had to bear the premium. It is not surprising that most of those who bought the insurance, used it when they needed it. Socio-economic status of families/individuals has not been captured adequately. Only variable related to that aspect included in the study is of Ethnicity. But it has not been discussed what the results signify.

An example of a significant finding is that utilisation was greater for women than men. The discussion needs to focus on such findings.

The study does not shed much light on many crucial aspects Nepal’s insurance programme – who were more likely to enroll than others, what kind of facilities were accessed and by whom, how did utilisation vary across type of diseases/conditions, what was the difference in out of pocket spending etc. A limitations section is needed.

5. The main finding in the qualitative analysis is the discrimination by hospitals between the insured and non-insured. Another important finding is that the poor were not yet given the waiver on premiums. The authors should discuss these aspects in relation to experience in other schemes in various countries.

6. The discussion compares figures found in the study with other countries without bothering to check whether the schemes are comparable at all. It is not very useful in its present form. Any comparisons should with schemes with similar design.

7. The manuscript should mention that this was a descriptive study and no comparison was done.

8. The language needs to be improved throughout the manuscript.

6. PLOS authors have the option to publish the peer review history of their article (what does this mean?). If published, this will include your full peer review and any attached files.

Reviewer #1: **Yes: **Sakthivel Selvaraj

Reviewer #2: **Yes: **Samir Garg

---

## [Author Response · Author response to Decision Letter 0]

15 Dec 2020

Author’s response to Reviewers’

Reviewer #1

The Govt. of Nepal has been implementing a Social Health Insurance program, and this particular research aims to assess utilization patterns of the scheme in one district, Illam, Nepal. While it is critical to conduct such exercise given its importance of being used as inputs for monitoring and revising the scheme design, following observations may be useful in respect to the manuscript in its current form.

The manuscript hardly provides any idea about the background of social security program. How many people are covered, which sections of population are covered, are there targeted mechanism? What is the benefit package offered by the scheme, and is this scheme purely health security or a larger social security scheme? and if so, what aspect of the survey captured in the manuscript? What kind of purchasing mechanism did the insurance scheme implement – did they empanel both public and private hospitals? A reading of the text later suggests that the manuscript deals with social health security and not social security scheme at large. Moreover, further down the text in sections dealing with qualitative results, the authors appear to use the term social health insurance. The manuscript must therefore clearly articulate what type of scheme are they referring to?

 Thank You for your comment. 

The background of social health security program has been provided from page 3, line 55 to page 4, line 81 as: 

Nepal remains committed to achieving universal health coverage, as seen in the new constitution, which was promulgated in 2015. [2] Although Nepal has a wide network of health facilities, only two-third of population has easy access to health care. [6] Despite various treatment subsidies like incentives program and pilot insurance scheme, out-of-pocket payment (OOP) is still the main way to finance health care which accounts for 48% of total health expenditure. [7] Thus, there was a felt need for a mechanism in the health financing system to enhance prepayment and risk pooling to minimise financial hardship and impoverishment arising from the use of health care services in Nepal. The government of Nepal introduced social health insurance Program as social health security Program in 2016 by rolling out in three districts Kailali, Baglung and Illam as piloting districts. The Social Health Security Programme (SHSP) is a social protection programme of the Government of Nepal that aims to enable its citizens to access quality health care services without placing a financial burden on them. In 2015 Government of Nepal established a semi-autonomous body, Social Health Security Development Committee which is now known as Health Insurance Board as an implementer of program which follows the structure of centre, province, district and ward. For the yearly membership fee (Premium) of Rs. 2500, an insured family of upto 5 member receives insurance card and an annual coverage of benefit package up to Rs. 50000 health related expenses in both public and private social health insurance empanelled hospitals. The scheme include the benefit package of emergency services, out-patient services, selected inpatient services, selected diagnostic services and selected drugs, in addition to any free services and drugs available at public health facilities through other programmes. The government provides subsidy to poor i.e. for ultra-poor 100%, Poor 75% and vulnerable groups 50% discount in premium. The choice of enrolment is voluntary. This cashless scheme covers the pre-existing condition and has no age limits. [2] The coverage rate is 5% covering 2, 38,344 people among the 15 implemented districts. [17] 

Study design:

 Why was Illam district selected as against other districts? ON what basis the samples were selected? Is that proportionate to social security coverage? The sample size calculation is utterly confusing, the method is unclear. A sample of 300 was collected based on Inpatient utilization of 38.5% (this is unrealistic given that more than one of the population of the district requires hospitalization in a year), which is usually 4-6% in most LMIC countries?

Thank You for the comment. 

 The reason for the selection of the Illam district against other districts has been mentioned in page no 6, line no 114-116 as 

Government of Nepal initiated the Social Health Security program in 2016 AD and piloted in three districts (Kailali, Baglung and Illam) from which Illam district was selected randomly through lottery method.

 The samples were selected based on their insured status and activation of service i.e. those households were selected who were insured and whose date of service has come into effect/activated. This has been mentioned in page no 6, line no 127-128

 Yes, that is proportionate to social security coverage. For this supporting information has been uploaded as “Sampling Procedure” as in page no 6, line no 123-124

 The method of sample size has been made clear in page no…. and line no…. as

A sample size of 300 was calculated using a formula n=〖(z〗^2 pq)/e^2 considering the prevalence of In-patient service utilization of 38.5% from the comparative study done in 2016 by Philip NE. et al. [13] which is done as follow: (page 5, line no 97-109)

z = 1.96 (95.0% Confidence level)

Prevalence of In-patient service utilization (p) = 38.5% [13]

q= 100-38.5 =61.5 %

e (margin error)= 15% of p = 15% of 0.385= 0.05775

Sample size (n) = 〖(z〗^2 pq)/e^2

 n = 1.96*1.96*0.385*0.615 / (.05775)2

 = 273.03

 n =273

Non response rate: 10.0% 

Total sample size = 300

 Yes, a sample of 300 was collected based on In-patient service utilization of 38.5% which is taken from literature of reference number [13] a study done in India as in page 5, line no 99

 Why choose six months rather than one year as data period. If this is for hospitalization did the survey collect information of self-reports for one year for hospitalization or 15/30 days for OP? The choice of recall period for OP and IP is not spelt out in the manuscript.

Thank You for the comment. 

The study was done for the partial fulfilment of Master in Public Health and we were provided the time frame of six months only. The survey collects information only on whether the sample has utilized the In-patient service (IP) or Out-patient services (OP) and other schemes, not the time period of utilization so recall period of OP and IP is not spelt out in manuscript.

 The method of selection of household was even more bizarre. Not sure what spinning the pen techniques is all about? A reference to this technique must be provided, if such a technique is in vogue or were practiced earlier?

Thank You for the comment. 

Spin the pen technique in perhaps the most common implementation; survey teams select a random starting direction from a central location in the cluster by spinning a pen or bottle. Households lying on this transect from the center to the border of the cluster are counted and one of them is then chosen at random. Proximity selection is then used to select subsequent households as the "next nearest" until the desired sample size is reached. More detail on spin the pen technique is given in literature with reference number [18].

 The qualitative method – no information is available in the text about what sets of variables are part of the questionnaire. Besides socio-economic profile of households, what other information were obtained? Did the tools collect information about different illness, did they record utilization of services by different types of providers, and did the researcher collect information about households’ OOP and if so, what are the key variables involved? None of these information/details are provided in the paper.

Thank You for the comment. 

The social health related knowledge, experience of getting enrolled in social health insurance and experience on utilization of social health insurance services are the sets of variable that were the part of the questionnaire in qualitative method. This has been included in page no 13, line no 226 to page no 16, line no 282.

The tool did not collect the information about different illness, did not record the utilization of service by different types of providers, and nor collect the information about households’ OOP. The key variable involved were the Knowledge, experience on getting enrolled and experience on health insurance service utilization. Please find the find the “focus group discussion guideline” as supporting information for more detail on variable for the qualitative part.

 From qualitative method, it was clear that the research was conducted only among those who were insured, leaving the uninsured out. It is equally possible that those who are eligible for insurance but uncovered for any reasons. And this set of population is expected to be significant, which is left out of the survey.

Thank You for the comment. 

Yes, the uninsured population has been left and it has been mention in limitation of the study. Page no 19, line no 368-371

Data Analysis:

The data analysis section is presented in a vague manner for any meaningful understanding? Why were binary logistic regression analysis chosen. What is the rationale behind choosing this analysis technique? What does the manuscript wants to convey when it says it carried out thematic analysis underlying qualitative data?

Thank You for the Comment

Binary logistic regression analysis was chosen because the dependent variable is dichotomous/binary i.e utilization of social health insurance which is dependent variable has binary response of “Yes” and “No”. Further detail on the binary logistic regression is given in reference [19] page 7, line no 157

The manuscript wants to convey that for the analysis of the qualitative data thematic analysis method was used. A thematic analysis is one that looks across all the data to identify the common issues that recur, and identify the main themes that summarise all the views you have collected. This is the most common method for descriptive qualitative study. This approach is useful for analysing responses to open-ended questions which was used in this study as “focus group discussion guidelines” (Attached as Supporting Information). For more detail on thematic analysis kindly refer to reference [21]

Findings:

- For readers in other parts of the world may not be familiar with words such as Brahmin/Chhetri reflecting ethnicity in Nepal. Perhaps a note on these characteristics is required.

- It is unrealistic to expect nearly half of the population (46.7%) to suffer from chronic illness. Either the method adopted to capture illness (self-reports) or the sample technique adopted was skewed. Even assuming nearly half the population have chronic illness in a particular district, does that reflect national average, or global evidence?

- While chronic illness were present in 46.7% of households as reported in Table 1, but

also, it appears that about 10.4% are reported to be hospitalized, which looks extremely high.

- How did the survey go about classifying the disease conditions (broader disease classification) as given in Table 2. Is that based on ICD-10 or what kind of classification?

- The section on qualitative data dealing with social health insurance related knowledge, it is abundantly clear that the scheme is yet to be implemented (since the poverty card in Illam district has not been distributed yet). In the absence of implementation of the scheme in the sample district, it is unclear what kind of results can be expected from the survey? But the next paragraph provides a contradictory view of beneficiaries receiving insurance benefits.

Thank You for the Comment

 Brahmin/Chhetri reflects upper caste people in society which is noted in page no 9, line 179 as “*Upper caste people in society”

 Nearly half of the population (46.7%) suffered from chronic illness. This might be so because the scheme is voluntary and covers pre-existing conditions too. This led to consumer side moral hazard i.e. those household which has pre-existing condition of chronic illness were more likely to get enrolled in scheme which is significant in this study. A study from India has also shown that the insured households had more number of people with chronic diseases, suggesting adverse selection from the point of view of an insurer, an expected phenomenon in a voluntary health insurance scheme. [13] Moreover going through the national data on chronic illness, according to Department of Health Services annual report 2017/2018 the deaths due to NCDs (cardiovascular disease, diabetes, cancer and respiratory disease) have increased from 60% of all deaths in 2014 to 66% in 2018 (WHO Nepal Country profile 2018).

 This study shows 10.4% of hospitalization (In-patient service utilization) which is not extremely high when comparing with others findings. A study from Kenya. India and China reported 20%, 38.5% 87.4% inpatient service utilization. [12, 13, 14] 

 The question regarding the disease condition was open ended and later it was converted in closed end by classifying the disease based on the body system e.g Pneumonia to respiratory system, Hypertension cardiovascular system.

 “Most of them were unaware about subsidize provided by social health insurance board for poor, ultra-poor and marginalized people. This is because health insurance board was unable to implement the policy as government has not distributed the poverty card in Illam district” Here this does not mean that scheme is yet to be implemented, what it means is the one of the policy that government has released regarding the subsidy to be provided on insurance premium for the Poor, ultra-poor and vulnerable group based on poverty card is not provides as government has not distributed the poverty card. It does not mean to say that scheme/social health insurance program is not implementing in illam district.

Reviewer #2

1. The insurance scheme details have not been provided – what are its stated objectives, what kind of people are eligible for insurance scheme, who pays the premium, what part of the population is eligible for waivers, what is the role of government, what kind of providers are used – public, private or both, what kind of care is included – inpatient/out-patient/both etc. If it is a voluntary insurance scheme, it should be mentioned upfront and compared with literature on voluntary insurance.

The insurance scheme details have been included in page no 3, line no 63-80 as 

The government of Nepal introduced social health insurance Program as social health security Program in 2016 by rolling out in three districts Kailali, Baglung and Illam as piloting districts. The Social Health Security Programme (SHSP) is a social protection programme of the Government of Nepal that aims to enable its citizens to access quality health care services without placing a financial burden on them. In 2015 Government of Nepal established a semi-autonomous body, Social Health Security Development Committee which is now known as Health Insurance Board as an implementer of program which follows the structure of centre, province, district and ward. For the yearly membership fee (Premium) of Rs. 2500, an insured family of upto 5 member receives insurance card and an annual coverage of benefit package up to Rs. 50000 health related expenses in both public and private social health insurance empanelled hospitals. The scheme include the benefit package of emergency services, out-patient services, selected inpatient services, selected diagnostic services and selected drugs, in addition to any free services and drugs available at public health facilities through other programmes. The government provides subsidy to poor i.e. for ultra-poor 100%, Poor 75% and vulnerable groups 50% discount in premium. The choice of enrolment is voluntary. This cashless scheme covers the pre-existing condition and has no age limits. [2]

2. Any literature available on functioning of the scheme and its origin should be included.

The literature on functioning of scheme and its origin is included as reference [2]

(Social Health Security Program: Standard Operating Procedure. Kathmandu: Government of Nepal Social Health Security Development Committee; 2016. 1-98.)

3. It is not clear why the scheme was introduced. Though the introduction mentions that the context was of moving away from public sector provisioning, it is not backed by any references.

The reason behind introducing the scheme has been explained in page no 3, line no 55-63 as 

Nepal remains committed to achieving universal health coverage, as seen in the new Constitution, which was promulgated in 2015. [2] Although Nepal has a wide network of health facilities, only two-third of population has easy access to health care. [6] Despite various treatment subsidies like incentives program and pilot insurance scheme, out-of-pocket payment (OOP) is still the main way to finance health care which accounts for 48% of total health expenditure. [7] Thus, there was a felt need for a mechanism in the health financing system to enhance prepayment and risk pooling to minimise financial hardship and impoverishment arising from the use of health care services in Nepal.

4. The main aspect the study is able to look at is utilisation of insurance scheme by those who were insured. It seems that the families had to bear the premium. It is not surprising that most of those who bought the insurance, used it when they needed it. Socio-economic status of families/individuals has not been captured adequately. Only variable related to that aspect included in the study is of Ethnicity. But it has not been discussed what the results signify.

An example of a significant finding is that utilisation was greater for women than men. The discussion needs to focus on such findings.

The study does not shed much light on many crucial aspects Nepal’s insurance programme – who were more likely to enroll than others, what kind of facilities were accessed and by whom, how did utilisation vary across type of diseases/conditions, what was the difference in out of pocket spending etc. A limitations section is needed.

The socio-economic status of families is included in findings table 1 as Poverty line of household while those variable with p-value <0.2 in bivariate analysis were only included for the binary logistic regression analysis. Therefore, Poverty line of household was not included for the binary logistic regression as its p-value was 0.253 and not a significant variable for the utilization of scheme. (page 9, line 176-179)

The study has made discussion on the significant finding from Page no 17, line 316 to page no 18, line no 338 as 

On the contrary to the study from Kerala India, in this study household having under five children were more likely to utilize the social health security scheme with odds of 2.915 (1.324-6.419) than household not having under five children. [13]

The study suggests that households having chronic illness were 3.830 (1.422-10.316) times more likely to utilize social health security scheme. This finding is consistent with study done in India which showed that presence of chronic illness in family was significantly, 2.1 times more likely to use the health insurance. [13] This suggests an expected phenomenon of voluntary health insurance scheme that there is likely to be adverse selection from an insurer side.

Utilization of social health security scheme was statistically significant with the sex of household members in this study. Males were 0.563 (0.448-0.707) times less likely to utilize the social health security scheme than females. A discussion paper from three African countries presented that Senegal has similar finding to this study i.e. males were less likely to use the health insurance service than females but in other two countries South Africa and Kenya; sex made no difference on service utilization. [12] A study conducted in India showed sex has no role in utilization of health insurance service. [13]

Older age group i.e. 60 and above years of people were significantly more likely to use the social health security scheme than other age group. This finding has been supported by the study from China and South Africa where older population were more likely to use the health service through insurance than other age group. [14, 12] An opposite situation was found in Kenya and Senegal, there was no significant difference in use of health service by senior and non-senior population. [12]

Limitation section has been included in page no 19, line no 368-371. as

The study could not cover the uninsured population. Therefore who were more likely to enrol than others, reason or barrier for not enrolling, what kind of facilities accessed by whom or utilization of service by different provider, what is the difference in out of pocket spending remained uncovered. However, this can be examined in further studies.

5.The main finding in the qualitative analysis is the discrimination by hospitals between the insured and non-insured. Another important finding is that the poor were not yet given the waiver on premiums. The authors should discuss these aspects in relation to experience in other schemes in various countries.

The main finding of qualitative analysis has been discussed in page no 19, line no 361-366 as- the qualitative findings (FGD) from the study done in India shared that Rashtriya Swasthya Bima Yojana (National Health Insurance Scheme) the people were unaware about the what services are provided by the Scheme and from which hospital they can get the service. Moreover, insurance did not reached the poor people and discrimination was done based on political power and insurer were compel to make additional payment and pressurized to buy medicine from out of pocket. [9]

6. The discussion compares figures found in the study with other countries without bothering to check whether the schemes are comparable at all. It is not very useful in its present form. Any comparisons should with schemes with similar design.

The researcher has tried to compare the figures found in the study with comparable studies from various countries.

7.The manuscript should mention that this was a descriptive study and no comparison was done.

The manuscript has mentioned that this is a descriptive study and no comparison was done in page no 5 and line no 91 as “A descriptive cross-sectional method was adopted with mixed method approach”

8.The language needs to be improved throughout the manuscript.

Language has been improved with language check by colleagues.

---

## [Decision Letter · Decision Letter 1]

11 May 2021

PONE-D-20-13306R1

Utilization of social health security scheme among the households of Illam district, Nepal

PLOS ONE

Dear Dr. Shah,

Thank you for submitting your manuscript to PLOS ONE. After careful consideration, we feel that it has merit but does not fully meet PLOS ONE’s publication criteria as it currently stands. Therefore, we invite you to submit a revised version of the manuscript that addresses the points raised during the review process.

Although your revision was not well-received by the reviewers, I do feel that you should have another opportunity to revise this. As Reviewer 1 notes, the English is often stilted and ungrammatical, and you will be well-served to have this copy-edited by someone with native English fluency who can improve the style and diction. You should also keep in mind that you are writing for an international audience, and people may need some basic information. For example, you should translate the value of Nepalese currency into dollars or euros, and perhaps put that in the context of the median income in the district. You also need to better explain the Brahmin/Janajati/Dalit distinction -- it is not really about ethnicity, as you say, or rather it is more complicated than that.

I also think the reviewers are perhaps a bit confused about the study population and the term "utilization." If I have this correct, you need to clarify that the subject population consists 100% of people who are enrolled in the health insurance scheme; however, "utilization" means going to a participating hospital for covered services. Some people obtain services locally from non-participating providers, and so have not "utilized" the scheme. If that is not correct, you need to explain the terms accurately.

We look forward to receiving your revised manuscript.

Kind regards,

M Barton Laws

Academic Editor

PLOS ONE

Journal Requirements:

Additional Editor Comments (if provided):

Reviewers' comments:

Reviewer's Responses to Questions

**Comments to the Author**

1. If the authors have adequately addressed your comments raised in a previous round of review and you feel that this manuscript is now acceptable for publication, you may indicate that here to bypass the “Comments to the Author” section, enter your conflict of interest statement in the “Confidential to Editor” section, and submit your "Accept" recommendation.

Reviewer #1: (No Response)

Reviewer #2: (No Response)

2. Is the manuscript technically sound, and do the data support the conclusions?

Reviewer #1: No

Reviewer #2: No

3. Has the statistical analysis been performed appropriately and rigorously? 

Reviewer #1: No

Reviewer #2: No

4. Have the authors made all data underlying the findings in their manuscript fully available?

Reviewer #1: Yes

Reviewer #2: No

5. Is the manuscript presented in an intelligible fashion and written in standard English?

Reviewer #1: No

Reviewer #2: No

6. Review Comments to the Author

Reviewer #1: Manuscript Title: Utilisation of social health security scheme among the households of Illam district, Nepal

Manuscript Number: PONE-D-20-13306

The Govt. of Nepal has been implementing a Social Health Insurance program, and this particular research aims to assess utilization patterns of the scheme in one district, Illam, Nepal.

The original manuscript was reviewed by me earlier, and the current form of the manuscript still suffers from the following:

The choice of recall period for OP and IP is not spelt out in the manuscript. The response of the authors is unclear. It appears from the description that it is current utilisation and apparently did not seek information about utilisation in the past – a relatively longer period from hospitalisation and a shorter recall period for outpatient visits. And yet the current utilisation rate is surprisingly large and therefore unreal.

Concepts are still not clear. The description of utilisation rate of social health insurance is unclear. What is the numerator and denominator used in the calculation of utilisation rate? Similarly, how are underutilisation measured – please indicate the numerator and denominator.

The sample size calculation estimating 300 samples was apparently done based on the prevalence of In-patient service utilisation of 38.5%. It is highly unlikely that one in three insured persons are hospitalised at any given point in time. On the contrary, Table 2 provides information about in-patient utilisation at 10.4%.

The response from authors about chronic illness being high needs to be reconfirmed. It is equally likely that chronic disease conditions may predominate in a community and increasingly so. But the manuscript is unclear about whether percentage mentioned for chronic conditions are distribution of chronic conditions among the insured or the rate of chronic conditions? In its current form the manuscript gives an impression that the rate of chronic conditions among the entire population is 47%, implying that half of the insured population have chronic conditions. This again seems unreal and unscientific, highlighting the bias in sample calculation.

Finally, the manuscript needs to be edited by a professional editor. In its current form, the manuscript suffers from severe language issues, including grammatical mistakes, improper sentence formation, etc. One can find such instances almost at every paragraph, which requires a thorough overhaul of the language.

Reviewer #2: The authors have not addressed the concerns raised by the reviewer. The design for the quantitative analysis remains inadequate.

7. PLOS authors have the option to publish the peer review history of their article (what does this mean?). If published, this will include your full peer review and any attached files.

Reviewer #1: **Yes: **Sakthivel Selvaraj

Reviewer #2: No

---

## [Author Response · Author response to Decision Letter 1]

16 Oct 2021

Author’s response to reviewers

PONE-D-20-013306

Title: “Utilization of social health security scheme among the households of Illam district, Nepal”

Authors:

Sanjeeb Shah

Nilambar Jha

Vijay Kumar Khanal 

Gyanu Nepal Gurung 

Babita Sharma

Mausam Shrestha

Version: 2 Date: 16 October 2021

Author’s response to editor’

The needful correction has been made as per the comment received from the editor.

 The manuscript has been thoroughly edited for the language usage, spelling and grammar.

The name of Person who edited the manuscript is Astha Ramaiya, PhD Scholar

A copy of manuscript showing a change by highlighting them has been uploaded as supporting information file

A clean copy of edited manuscript has been uploaded as manuscript file

 Translation of Nepalese Currency into dollar has been done at page 3 line 66-68 that is Rs. 2500 to $22 and Rs. 50000 to $441 as per 1 USD= 113.38 NPR on 2019/01/18

 Regarding explanation of Brahmin/Janajati/Dalit distinction it has been noted in Page 9 line 171-172 as:

 Brahim/Chhetri-upper caste group, Janjati and Dalit Lower caste/Disadvantage group in society

 On study Population, It has been mentioned that subject population consist 100% of people who are enrolled in the health insurance scheme in Page 4 line 85-88 as:

 A descriptive cross-sectional method was adopted with mixed method approach to study the 300 households that were enrolled in the health insurance scheme of Illam district of Nepal between September 2018 – February 2019 and whose date of service had come into effect.

 Regarding confusion on term ” Utilization” explanation of Utilization of scheme has been mentioned in section operational definition, page 20 line 383-386 as:

Whenever the beneficiary had used the insurance card to use the benefit package or taken the benefits of social health insurance scheme, it will be considered as” utilization of scheme” 

Author’s response to Reviewers’

Reviewer #1

The choice of recall period for OP and IP is not spelt out in the manuscript. The response of the authors is unclear. It appears from the description that it is current utilisation and apparently did not seek information about utilisation in the past – a relatively longer period from hospitalisation and a shorter recall period for outpatient visits. And yet the current utilisation rate is surprisingly large and therefore unreal.

-The choice of recall period for OP and IP and other schemes is spelt out in the manuscript in Data collection section in page 6, line no 132-134 as:

Information on utilization of In-patient and Out-patient and other schemes was obtained for 12 months before the date of interview.

Concepts are still not clear. The description of utilisation rate of social health insurance is unclear. What is the numerator and denominator used in the calculation of utilisation rate? Similarly, how are underutilisation measured – please indicate the numerator and denominator.

 Here Utilization of social health insurance is described in section operational definition, page 20 line 383-386 as:

Whenever the beneficiary had used the insurance card to use the benefit package or taken the benefits of social health insurance scheme, it will be considered as” utilization of scheme” 

 The numerator and denominator used in the calculation of utilisation rate are “Household utilized social health insurance” and “Total Sample Size” respectively.

Utilization rate=(Household utilized social health insurance÷Total Sample size)×100 

 =(266÷300)×100

 = 88.7

 The numerator and denominator used to measure underutilisation are “Household did not utilize social health insurance” and “Total Sample size” respectively.

Underutilization rate=(Household did not utilize social health insurance÷Total Smaple size)×100 

 =(34÷300)×100

 = 11.3

The sample size calculation estimating 300 samples was apparently done based on the prevalence of In-patient service utilisation of 38.5%. It is highly unlikely that one in three insured persons are hospitalised at any given point in time. On the contrary, Table 2 provides information about in-patient utilisation at 10.4%.

 Yes from literature of reference number [13] a study done in India as in page 5, line no 99, a sample of 300 was collected based on In-patient service utilization of 38.5% which is taken 

The reason for high In-patient utilization in the study done in India is that the Comprehensive health insurance scheme covered only Inpatient care which lead to increase in inpatient utilization (Mentioned in article discussion section, Philip NE, Kannan S, Sharma SP. Utilization of Comprehensive Health Insurance Scheme, Kerala: A Comparative Study of Insured and Uninsured Below-Poverty-Line Households. Asia-Pacific Journal of Public Health 2016; 28: 77-85.)

The response from authors about chronic illness being high needs to be reconfirmed. It is equally likely that chronic disease conditions may predominate in a community and increasingly so. But the manuscript is unclear about whether percentage mentioned for chronic conditions are distribution of chronic conditions among the insured or the rate of chronic conditions? In its current form the manuscript gives an impression that the rate of chronic conditions among the entire population is 47%, implying that half of the insured population have chronic conditions. This again seems unreal and unscientific, highlighting the bias in sample calculation

The percentage mentioned for chronic condition are distribution of chronic conditions among the insured sampled household (that is out of 300 hundred insured sampled household 140 household had chronic illness 46.7%). It’s not the rate of chronic condition among the entire population.

This might be so because the scheme is voluntary and covers pre-existing conditions too. This led to consumer side moral hazard i.e. those household which has pre-existing condition of chronic illness were more likely to get enrolled in scheme which is significant in this study.

Finally, the manuscript needs to be edited by a professional editor. In its current form, the manuscript suffers from severe language issues, including grammatical mistakes, improper sentence formation, etc. One can find such instances almost at every paragraph, which requires a thorough overhaul of the language.

The manuscript was edited by Astha Ramaiya, PhD Scholar to solve the language issues, including grammatical mistakes and improper sentence formation.

Reviewer #2

The authors have not addressed the concerns raised by the reviewer. The design for the quantitative analysis remains inadequate.

The author have addressed the concern raised by the reviewer which is highlighted in manuscript

 The design for the quantitative analysis has been explained in page no. 7 line 142-150 as: Collected data was entered in Microsoft Excel 2010 and was analyzed in Statistical Package for Social Science 11.5. Descriptive, bivariate and multivariate analysis was done. Bivariate analysis was done using chi-square test to establish relationship between dependent (utilization of social health security scheme) and independent (household characteristics) variables. The level of significance was set at 5%. Variables with p-value<0.2 and cell count more than 5 in the bivariate analysis were included for the binary logistic regression analysis to find the factors associated with utilization of social health security scheme. For qualitative data thematic analysis was conducted

Binary logistic regression analysis was chosen because the dependent variable is dichotomous/binary i.e utilization of social health insurance which is dependent variable has binary response of “Yes” and “No”. Further detail on the binary logistic regression is given in reference [19] page 23, line no 443.

---

## [Editor Report · Decision Letter 2]

20 Oct 2021

PONE-D-20-13306R2Utilization of social health security scheme among the households of Illam district, NepalPLOS ONE

Dear Dr. Shah,

Thank you for submitting your manuscript to PLOS ONE. After careful consideration, we feel that it has merit but does not fully meet PLOS ONE’s publication criteria as it currently stands. Therefore, we invite you to submit a revised version of the manuscript that addresses the points raised during the review process.

Rather than send this out again for peer review, I think it will be most helpful at this stage if I provide specific comments of my own. First, while the English is still somewhat stilted and contains many grammatical errors, it is at least now largely intelligible. I would still recommend that you get another round of copy editing from someone fluent in English, but there are more important problems of organization and clarity.

These problems begin with the abstract, where you write "A . . . study was conducted among 300 households of Illam district which had utilized the social health security scheme." This is, of course, incorrect. The point of the study was to find out whether or not people who are enrolled in the scheme had utilized it, and you found that indeed some had not. In the text, you do say that eligible households are those that had enrolled in the scheme, but the error in the abstract will confuse readers. You do need to explain in the introduction that the service benefit is available only through certain facilities. It appears these are government-run medical centers that provide both inpatient and outpatient services, but you need to clearly explain the nature of these facilities and how accessible they are to the population.

The power analysis is probably unnecessary. I will say, however, that the prior you use is, as you explain in your response to reviewer, evidently invalid. It assumes very high inpatient utilization of 38.5%, which apparently results from moral hazard in the comparator. But you can just delete this. You are not asking for funding, but presenting results.

Regarding the "spin the pen" technique for selecting households, evidently you had maps available showing all of the enrolled households in the district. You should explain how these were generated.

As for reporting of the qualitative component, we need much more detail. One way to assure that you have reported your methods adequately is to use the COREQ checklist. Not every element of it necessarily applies to this study, but if you try to respond to most of the items reviewers will have a better idea of what you actually did. PDF versions are available free on-line. We need much more information about how respondents were recruited, how the discussion guide was developed, how the focus groups were conducted and how the data was analyzed.

You should explain the ethnic categories in the text, not just in the table note. You should explain what you mean by "nuclear family." I presume you mean parents and children only, whereas other households are extended families including other relatives? Are they all eligible for enrollment in the scheme as a single unit? Presumably people who had not utilized the scheme because the nearby private facility was easy to access paid out of pocket. If you had explained earlier that the service benefit could only be accessed at public facilities this would be easier to understand at this point. When you introduce the concept of "minor morbidity" at line 180 you need to define it.

Much of what you have placed under the heading of "discussion" actually consists of findings that you had not previously reported. All of your observations must be placed in the findings. The discussion is where you can provide your recommendations. I think the comparison with other countries can be greatly condensed; much of it is not very informative.

If you can respond to these comments, I can consider whether the paper is acceptable as revised, or needs to be sent out again for peer review. But peer review will be much more constructive if you can make these revisions first.

We look forward to receiving your revised manuscript.

Kind regards,

M Barton Laws

Academic Editor

PLOS ONE

Additional Editor Comments (if provided):

Rather than send this out again for peer review, I think it will be most helpful at this stage if I provide specific comments of my own. First, while the English is still somewhat stilted and contains many grammatical errors, it is at least now largely intelligible. I would still recommend that you get another round of copy editing from someone fluent in English, but there are more important problems of organization and clarity.

These problems begin with the abstract, where you write "A . . . study was conducted among 300 households of Illam district which had utilized the social health security scheme." This is, of course, incorrect. The point of the study was to find out whether or not people who are enrolled in the scheme had utilized it, and you found that indeed some had not. In the text, you do say that eligible households are those that had enrolled in the scheme, but the error in the abstract will confuse readers. You do need to explain in the introduction that the service benefit is available only through certain facilities. It appears these are government-run medical centers that provide both inpatient and outpatient services, but you need to clearly explain the nature of these facilities and how accessible they are to the population.

The power analysis is probably unnecessary. I will say, however, that the prior you use is, as you explain in your response to reviewer, evidently invalid. It assumes very high inpatient utilization of 38.5%, which apparently results from moral hazard in the comparator. But you can just delete this. You are not asking for funding, but presenting results.

Regarding the "spin the pen" technique for selecting households, evidently you had maps available showing all of the enrolled households in the district. You should explain how these were generated.

As for reporting of the qualitative component, we need much more detail. One way to assure that you have reported your methods adequately is to use the COREQ checklist. Not every element of it necessarily applies to this study, but if you try to respond to most of the items reviewers will have a better idea of what you actually did. PDF versions are available free on-line. We need much more information about how respondents were recruited, how the discussion guide was developed, how the focus groups were conducted and how the data was analyzed.

You should explain the ethnic categories in the text, not just in the table note. You should explain what you mean by "nuclear family." I presume you mean parents and children only, whereas other households are extended families including other relatives? Are they all eligible for enrollment in the scheme as a single unit? Presumably people who had not utilized the scheme because the nearby private facility was easy to access paid out of pocket. If you had explained earlier that the service benefit could only be accessed at public facilities this would be easier to understand at this point. When you introduce the concept of "minor morbidity" at line 180 you need to define it.

Much of what you have placed under the heading of "discussion" actually consists of findings that you had not previously reported. All of your observations must be placed in the findings. The discussion is where you can provide your recommendations. I think the comparison with other countries can be greatly condensed; much of it is not very informative.

If you can respond to these comments, I can consider whether the paper is acceptable as revised, or needs to be sent out again for peer review. But peer review will be much more constructive if you can make these revisions first.
---

## [Author Response · Author response to Decision Letter 2]

10 Mar 2022

Author’s response to reviewers

PONE-D-20-13306R2

Title: “Utilization of social health security scheme among the households of Illam district, Nepal”

Authors:

Sanjeeb Shah

Nilambar Jha

Vijay Kumar Khanal 

Gyanu Nepal Gurung 

Babita Sharma

Mausam Shrestha

Version: 3 Date: 10 March 2022

Author’s response to editor’

The needful correction has been made as per the comment received from the editor.

 A statement in the abstract “A . . . study was conducted among 300 households of Illam district which had utilized the social health security scheme” has been changed at page no. 1 line no 18-20 as “A descriptive cross-sectional mixed-method study was conducted among 300 households of Illam district who had enrolled in the social health security scheme”.

 In introduction; the service benefit is available only through certain facilities has been explained at page no. 3 line no 59-62 as “the government of Nepal introduced the social health insurance program to government run health facility (Primary Health Care Centre, District Hospital which are reachable within one hours) that provide out-patient and in-patient service to mitigate the financial hardship due to healthcare and increase access to healthcare”.

 For the power analysis; it has changed as follow at page no. 5 line no 95-97 as “A sample size of 300 was calculated using a formula n=〖(z〗^2 pq)/e^2 considering the prevalence from study done in 2016 by Philip NE. et al. in Kerala India and non-response rate of 10%”.

 Regarding the map generation for “spin the pen”, explanation has been provided at page no. 6 line no. 118-120 as “For this map was generated manually with support of social health insurance supervisor of Illam district based on data available there”. 

 Reporting for qualitative component, based on COREQ checklist information has been provided at page no. 7 line no. 130-139 & 150-151 as “The principle of phenomenology was used to conduct three focus group discussions (FGD) in selected three municipalities (one in each municipality) using pretested focus group guideline prepared based on the theme. Each focus group discussion lasted for 30-40 minutes which included 10 participants chose purposively from insured household, whose date of service was into effect, ensuring representation of gender, different castes, religious group and elderly. Note taking as well as audio recording of discussion was done after taking consent from participants. The purpose of FGD was to record the community’s perception of social health security scheme.

For qualitative, data was coded, transcribed and thematic analysis was performed manually where themes were identified in advance.”

COREQ checklist information has been uploaded as supplementary document too

 Ethnic categories and family type has been explained in text at page no. 8 line no. 165-169 as Brahmin/Chhetri are considered to be the upper caste group, Janjati and Dalit are considered to be the lower caste/disadvantage group in society and nuclear families comprises of parents and children whereas other households were extended families including other relatives too.

Regarding the eligibility for enrollment in the scheme as a single unit and accessibility of service benefit has been mentioned at page no. 4 line no. 69-72 as “For a yearly membership fee (Premium) of Rs. 2500 ($22), a family of upto five member could receive an insurance card and an annual coverage of benefit package up to Rs. 50000 ($441) health related expenses in both public and private social health insurance empanelled hospitals”

The concept of “minor morbidity” has been defined at page no. 20 line no. 370-371 as “Any morbidity of sudden onset which affected the activities of daily living for more than 24 hours.”

 The discussion part has been modified as per provided comments at page no. 16-19 line no. 281-353 as 

“Utilization of Social Health Security Scheme

Limited studies have been conducted on the social health insurance program in the context of Nepal. The study revealed higher utilization of social health security scheme which is consistent with the studies done in Philippines and Rwanda. [8, 11] The utilization of outpatient service was higher than inpatient service which aligned with the finding from South Africa, Kenya, Senegal and contrast to India and China. [12, 13, 14] The underutilization of the scheme was slightly lower than that of study from Philippines. [8] The reason for this might be due to easy access to private health services.

Factors associated with utilization of social health security scheme

The study depicted that households having under five children and chronic illness were more likely to utilize the social health security scheme, similar to India’s finding [13] suggesting expected phenomenon of voluntary health insurance scheme that there is likely to be adverse selection from an insurer side. In contrast, to study done in South Africa, Kenya and India sex has made difference on scheme utilization in this study. Regarding age, older age group (i.e. 60 and above years of people were more likely to utilize the social health security scheme which is similar with to finding from South Africa and China. [12, 14]

Community’s Perception

The FGD enlightened that people had understood what social health insurance program is all about. They came to know about it through radio, television, FCHVs and enrollment assistant. However, none of them exactly knew about what services are provided through social health insurance in hospital and most of them were unaware about the subsidize provided by social health insurance board for poor, ultra-poor and marginalized people as the insurance board was unable to implement the policy because government has not distributed the poverty card in Illam district. 

Irrespective of people different financial status, they had enrolled in social health insurance because they had the concept that health insurance protects their family from unpredicted financial risk when they get health problem. In addition, it was easy for people to get enrolled in social health insurance because for each ward one enrollment assistant has been appointed.

People supported the social health insurance program but they had unpleasant experience while taking service at hospital. They faced discrimination with uninsured people, misbehaviors at hospital pharmacy, have to go through numbers of registration counters, and did not find the doctors, all kind of lab services, drugs, and difficulty with referral system. Therefore, for the further improvement of the social health insurance program, people suggested to treat everyone equally in hospital, all the services should be available in nearby hospital, should be able to use insurance card at any place, at any time, at any hospital under social health insurance without going through referral procedure. The qualitative findings (FGD) from the study done in India shared that Rashtriya Swasthya Bima Yojana (National Health Insurance Scheme) the people were unaware about the what services are provided by the Scheme and from which hospital they can get the service. Moreover, insurance did not reached the poor people and discrimination was done based on political power and insurer were compel to make additional payment and pressurized to buy medicine from out of pocket. [9]

Limitations

There might have occurred recall bias as the recall period of utilization of service was 12 month before the interview. However, it was minimized with medical bills and report wherever provided but was not applicable in each household. The study could not cover the uninsured population. Therefore who were more likely to enrol than others, reason or barrier for not enrolling, what kind of facilities accessed by whom or utilization of service by different provider, what is the difference in out of pocket spending remained uncovered. However this can be examined in further studies.

Conclusion and Recommendation

In Nepal with increased focus on achieving universal health coverage through increasing access to health care, the study provides intuition to which extent this dimension of universal health coverage has been achieved through social health security scheme program in Illam district. Although the utilization of social health security scheme was higher and had positive response towards the program, people were unsatisfied with the service and facility provided by the hospital under the social health security scheme. Therefore, the investment of demand-side will not be justified unless the basic health care infrastructures are upgraded to specialist health besides serving as gatekeeping for specialist service. In addition to that the concern authority should invest in system to monitor the hospitals and develop a feedback mechanism from users so to evaluate the implementation of health insurance scheme. This will further help the government to refine the policy, plan and strategies in addressing the factors that hindered the consumers’ satisfaction. Moreover, such system will provide robust path in achieving the government’s goal of scaling up the program nationwide. Further, a study from provider perspective would give more clear insight on implementation situation of social health security program.”

 Translation of Nepalese Currency into dollar has been done at page 3 line 66-68 that is Rs. 2500 to $22 and Rs. 50000 to $441 as per 1 USD= 113.38 NPR on 2019/01/18

 On study Population, It has been mentioned that subject population consist 100% of people who are enrolled in the health insurance scheme in Page 4 line 85-88 as:

 A descriptive cross-sectional method was adopted with mixed method approach to study the 300 households that were enrolled in the health insurance scheme of Illam district of Nepal between September 2018 – February 2019 and whose date of service had come into effect.

 Regarding confusion on term ” Utilization” explanation of Utilization of scheme has been mentioned in section operational definition, page no.19- 20 line 363-366 as:

Whenever the beneficiary had used the insurance card to use the benefit package or taken the benefits of social health insurance scheme, it will be considered as” utilization of scheme” 

 A copy of manuscript showing a change by highlighting them has been uploaded as supporting information file

A clean copy of edited manuscript has been uploaded as manuscript file

Author’s response to Reviewers’

Reviewer #1

The choice of recall period for OP and IP is not spelt out in the manuscript. The response of the authors is unclear. It appears from the description that it is current utilisation and apparently did not seek information about utilisation in the past – a relatively longer period from hospitalisation and a shorter recall period for outpatient visits. And yet the current utilisation rate is surprisingly large and therefore unreal.

-The choice of recall period for OP and IP and other schemes is spelt out in the manuscript in Data collection section in page 6, line no 127-129 as:

Information on utilization of In-patient and Out-patient and other schemes was obtained for 12 months before the date of interview.

Concepts are still not clear. The description of utilisation rate of social health insurance is unclear. What is the numerator and denominator used in the calculation of utilisation rate? Similarly, how are underutilisation measured – please indicate the numerator and denominator.

 Here Utilization of social health insurance is described in section operational definition, page no. 19-20 line no. 363-366 as:

Whenever the beneficiary had used the insurance card to use the benefit package or taken the benefits of social health insurance scheme, it will be considered as” utilization of scheme” 

 The numerator and denominator used in the calculation of utilisation rate are “Household utilized social health insurance” and “Total Sample Size” respectively.

Utilization rate=(Household utilized social health insurance÷Total Sample size)×100 

 =(266÷300)×100

 = 88.7

 The numerator and denominator used to measure underutilisation are “Household did not utilize social health insurance” and “Total Sample size” respectively.

Underutilization rate=(Household did not utilize social health insurance÷Total Smaple size)×100 

 =(34÷300)×100

 = 11.3

The response from authors about chronic illness being high needs to be reconfirmed. It is equally likely that chronic disease conditions may predominate in a community and increasingly so. But the manuscript is unclear about whether percentage mentioned for chronic conditions are distribution of chronic conditions among the insured or the rate of chronic conditions? In its current form the manuscript gives an impression that the rate of chronic conditions among the entire population is 47%, implying that half of the insured population have chronic conditions. This again seems unreal and unscientific, highlighting the bias in sample calculation

The percentage mentioned for chronic condition are distribution of chronic conditions among the insured sampled household (that is out of 300 hundred insured sampled household 140 household had chronic illness 46.7%). It’s not the rate of chronic condition among the entire population.

This might be so because the scheme is voluntary and covers pre-existing conditions too. This led to consumer side moral hazard i.e. those household which has pre-existing condition of chronic illness were more likely to get enrolled in scheme which is significant in this study.

Reviewer #2

The authors have not addressed the concerns raised by the reviewer. The design for the quantitative analysis remains inadequate.

The author have addressed the concern raised by the reviewer which is highlighted in manuscript

 The design for the quantitative analysis has been explained in page no. 7 line 142-149 as: “Collected data was entered in Microsoft Excel 2010 and was analyzed in Statistical Package for Social Science 11.5. Descriptive, bivariate and multivariate analysis was done. Bivariate analysis was done using chi-square test to establish relationship between dependent (utilization of social health security scheme) and independent (household characteristics) variables. The level of significance was set at 5%. Variables with p-value<0.2 and cell count more than 5 in the bivariate analysis were included for the binary logistic regression analysis to find the factors associated with utilization of social health security scheme”. 

Binary logistic regression analysis was chosen because the dependent variable is dichotomous/binary i.e utilization of social health insurance which is dependent variable has binary response of “Yes” and “No”. Further detail on the binary logistic regression is given in reference [19] page 22, line no 423.

---

## [Editor Report · Decision Letter 3]

14 Mar 2022

Utilization of social health security scheme among the households of Illam district, Nepal

PONE-D-20-13306R3

Dear Dr. Shah,

We’re pleased to inform you that your manuscript has been judged scientifically suitable for publication and will be formally accepted for publication once it meets all outstanding technical requirements.

Kind regards,

M Barton Laws

Academic Editor

PLOS ONE

Additional Editor Comments (optional):

I believe you have responded adequately to the reviewers' comments and to mine. The English diction remains a problem, but the paper is intelligible, and the explanations of the context and the research questions, methods and results are now clear. I would still recommend that you have this copy edited by someone of native English fluency, but I find it acceptable for publication.
---

## [Editor Report · Acceptance letter]

19 Apr 2022

PONE-D-20-13306R3 

Utilization of social health security scheme among the households of Illam district, Nepal 

Dear Dr. Shah:

I'm pleased to inform you that your manuscript has been deemed suitable for publication in PLOS ONE. Congratulations! Your manuscript is now with our production department. 

Kind regards, 

on behalf of

Dr. M Barton Laws 

Academic Editor

PLOS ONE